

# Spatial variations, origins, and risk assessments of polycyclic aromatic hydrocarbons in French soils

Claire Froger[*a], Nicolas P. A. Saby[a], Claudy C. Jolivet[a], Line Boulonne[a], Giovanni Caria[b], Xavier Freulon[c],

5    Chantal de Fouquet[c], Hélène Roussel[d], Franck Marot[d], Antonio Bispo[a,d]

[a] INRAE, US1106 Unité Infosol, Centre de Recherches d'Orléans, CS 40001, Ardon, 45075 Orléans Cedex 2,
France

[b] INRAE, US0010 Laboratoire d'analyses des sols, 273 rue de Cambrai 62000 Arras, France

10    [c] Mines ParisTech, Centre de géosciences. 35, rue saint-Honoré. 77305 Fontainebleau, France. 75006 Paris,
France
[d] ADEME, 20 Ave Gresille, F-49004 Angers 01, France

*Corresponding author:* Claire Froger (claire.froger@inrae.fr)

15    **Abstract**. Polycyclic aromatic hydrocarbons (PAHs) are persistent organic pollutants produced by anthropogenic
activities that contaminate all environmental spheres, including soils. This study focused on PAHs measured in
2154 soils in France, covering the entire territory based on a regular sampling grid. The quantified concentrations
in the $\Sigma 15 PAHs$ ranged from 5.1 to 31200 $\mu g \cdot kg^{-1}$, with a median value of 32.6 $\mu g \cdot kg^{-1}$, and PAHs were detected
in 70% of the soil samples. The map of $\Sigma 15 PAHs$ concentrations revealed strong spatial variations in soil

contamination throughout France, with larger concentrations in soils of industrial regions and near major cities.
PAHs molecular diagnostic ratios supports the historical origin of PAHs in the northern part of France being linked
to the significant emissions of PAHs in Europe during the industrial period of 1850–1950 with in particular the
contribution of coal/biomass combustion and iron-steel production. A health risk assessment conducted for the
residential population resulted in a median value of $1.07 \times 10^{-8}$ in total lifetime cancer risk, with only 20 sites

above the limit of $10^{-6}$ and one above the limit of $10^{-5}$ adopted by the French government. These results reveal the
need to conduct large-scale studies on soil contamination to determine the fate of PAHs and evaluate the risks
induced by soil pollution at a country-level scale.

Keywords: PAHs; soil monitoring; geostatistics; molecular diagnostic ratios; health risk

## 1   Introduction

Polycyclic aromatic hydrocarbons (PAHs) have been targeted by worldwide environmental and public health
agencies since 1970 due to their mutagenicity and carcinogenicity (Grimmer, 1985). These substances, which are
emitted during the incomplete combustion and pyrolysis of organic matter, are mainly associated with
anthropogenic activities, especially 'heavy' industries (e.g. steel and coal), residential heating, and road traffic
(Pacyna et al., 2003). Due to more than 150 years of contaminant release via human activities, all environmental

spheres are now affected by persistent organic contaminants, including PAHs. Consequently, European measures
on pollution control have been implemented, such as the Water Framework Directive or the Convention on Long-
Range Transboundary Air Pollution; however, no such directives exist for soil quality, despite the fact that soil

contamination is a key challenge in the worldwide management of contaminants to protect both human health and ecosystem quality (Carre et al., 2017). The various studies that can be found in the literature on PAH soil

contamination mainly focus on local soil pollution (Liu et al., 2016; Yang et al., 2008) or on soils under a particular type of land use (Cachada et al., 2012a; Jensen et al., 2007; Wang et al., 2017a). To efficiently manage soil contamination and prevent the release of pollutants stored in soils, we need to work at the scale of countries to determine the spatial variations in these contaminants and evaluate their risks to both ecosystems and human populations.

To fully understand the spatial distribution of PAHs, especially at a national scale, we need to determine the origin of the contaminants to potentially enact mitigation techniques at their sources. Considering the widespread contamination of Western Europe by the PAHs released since 1850 (Fernández et al., 2000), it is necessary to determine whether the contamination of soils observed nowadays is the result of legacy pollution or the current release of these substances. Multiple tracing methods have been developed to identify PAH sources. Among these

techniques, PAH molecular ratios have been widely used to discriminate PAHs according to their pyrogenic and petrogenic origins (Ravindra et al., 2008; Yunker et al., 2002); however, these PAH ratios appear to be sensitive to environmental processes such as oxidation and photo-oxidation, resulting in their modification during PAH transport due to the preferential degradation of some PAHs (Biache et al., 2014; Zhang et al., 2005). To prevent misinterpretations, the use of a combination of multiple PAH ratios is necessary to more accurately identify the

origin of PAHs (Kavouras et al., 2001; Tobiszewski and Namieśnik, 2012).

Due to health issues that are attributed to PAHs, an assessment of risks posed by contaminated soil to humans needs to be conducted to proceed with land management decisions. The United States Environmental Protection Agency (US EPA) developed guidelines to conduct human health risk assessments prior to the remediation of contaminated sites (US EPA 1991), and they then created a manual for soil screening (US EPA 1996).

Improvements were made to adapt the risk assessment to the various types of soil contaminant exposure, including residential, occupational, and recreational exposure (US EPA 1996, 2020). In addition, the parameters used in the risk calculations are regularly updated based on the results reported in the literature (IRIS et al., 2017). Therefore, a health risk assessment has been applied in multiple studies addressing PAHs in soils (Albanese et al., 2015; Cachada et al., 2016; Wang et al., 2017b) using a determinist point estimate approach to first evaluate the health

risks posed by soils, as recommended by the US EPA (2001).

Only a handful of studies have examined PAHs in soils at national scales such as those in the United Kingdom (Heywood et al., 2006), China (Ma et al., 2015), Switzerland (Desaules et al., 2010), Germany (Aichner et al. 2013), and the Netherlands (Brus et al., 2009). Unfortunately, the selected sampling design (e.g. Brus et al. 2009), soil sampling resolution, or land-use specifications (Aichner et al., 2013) have limited the use of these relevant

datasets for accurately mapping the PAH spatial distribution at national scales. Currently, only Villanneau et al. (2013) has attempted to map the distribution of a selection of PAHs in French soils based on a coarse sampling grid corresponding to a subset of 549 soils collected within the framework of the French Soil Quality Monitoring Network (RMQS: Réseau de Mesure de la Qualité des Sols). Their paper provided a first look at the spatial distribution of PAHs at the national scale; however, only four compounds (Benzo(b)fluoranthene, Fluoranthene,

Pyrene and Phenanthrene) met the criteria to be mapped using geostatistical technics. The four resulting maps showed a general trend of PAHs in French soils but only broad scale spatial variation were detected in PAHs contamination due to the low-resolution of the input data. As this first work gave limited insights either in the



identification of PAHs origins and their distribution, working with the sum of all PAHs molecules is therefore necessary to look further into PAHs distribution over the territory and to identify their origins.

Finally, of the large-scale studies of PAHs in soils that have been reported, the spatial distribution and contaminant levels have been the main focus, while health risk assessments and source identification have only been assessed at local scales (Morillo et al., 2008; Wang et al., 2015, 2013). Therefore, it is necessary to conduct complete studies at national scales, including the identification of major PAH sources to understand their spatial distribution and an evaluation of the associated risks to build effective national policies.

The objectives of the current study are thus: i) to evaluate the contamination of soils by PAHs in France in comparison with international soil monitoring studies; ii) to produce a high resolution map of the spatial distribution of Σ15PAHs in French soils; iii) to determine the main origins of PAHs in soils based on PAH molecular ratios; iv) to evaluate the potential health risks for the residential population associated with PAHs in French soils in order to have a first look at the risks induced by the background contamination of soils.


## 2    Materials and methods

### 2.1    Sampling sites

The sampling sites were selected based on a regular grid of 16 km × 16 km, and they were sampled at the centre of each cell or in a radius of less than 1 km from the theoretical centre when no soil was found at the predicted 95    location (Figure S1). Approximately 2200 sites were sampled, which consisted mostly of agricultural lands (50%), grasslands (20%), and forests (25%), with 1.2% of the region being urbanised areas (including brownfields) and 2.7% of the region being natural areas.

### 2.2    Sampling procedure and analysis

The sampling procedure and PAH analyses have been detailed in Villanneau et al. (2009; 2013); the soils were 100    sampled from June 2000 to May 2010 using a systematic procedure. At each site, in a 20 m × 20 m grid, 25 core samples of topsoil (approximately 0–30 cm) were taken. The core samples were bulked to obtain a composite sample, which was air-dried and sieved to 2 mm before analysis.

The concentrations of the 16 PAH compounds were measured in the topsoil composite samples: naphthalene (Naph), acenaphtylene (Acy), acenaphthene (Ace), fluorene (Flu), phenanthrene (Phe), anthracene (Ant), 105    fluoranthene (Flh), pyrene (Pyr), benzo(a)anthracene (BaA), chrysene (Chry), benzo(b)fluoranthene (BbF), benzo(k)fluoranthene (BkF), benzo(a)pyrene (BaP), dibenzo(ah)anthracene (DahA), indeno(123,cd)pyrene (IndP), and benzo(ghi)perylene (BghiP). These analyses were conducted at the Laboratory of Soils Analysis of the National Research Institute of Agronomy (INRA, in French) in Arras, France; this laboratory is accredited for PAH analysis in soils by the French accreditation committee (COFRAC). All the analyses were conducted between 110    2008 and 2012; consequently, volatilization of lighter PAHs such as Naph could have occurred during the long-term storage of soils collected at the beginning of the sampling campaign.

A subsample of ~20 g was taken from topsoil composites, and PAHs were extracted using a Dionex ASE 200 extractor to perform pressure liquid extraction at high pressure (103.4 bar) and temperature (150 °C) with a mixture of acetone/hexane/dichloromethane (50/25/25; v/v/v). After extraction, the samples were partially concentrated by



rotary evaporation to a volume of 5 mL and then completely evaporated under a gentle flow of nitrogen. The dry
residue was taken up with 2 mL of methanol and filtered through a 0.2 µm PTFE filter to remove solid particles,
which were finally transferred into 2 mL vials. The extracts were then injected into a thermoelectric high-
performance liquid chromatograph (HPLC) with a quaternary pump P4000, an autosampler AS3000, a commutator
SN4000 coupled with two Thermoelectron detectors mounted in series, a diode array ultraviolet (UV6000), and a

spectro-fluorimeter (F3000). A 10 µL–volume aliquot was injected in the HPLC, and the PAHs were separated on
a Hypersil PAH column (100 mm in length, 0.46 mm diameter, 5 µm particle size) of a C18 reversed phase. In-
depth technical details of the chromatographic parameters used are provided by Villanneau et al. (2009).

External calibration was used to quantify PAHs with concentrations from 40 to 1,000 µg·L$^{-1}$ detected by
fluorescence, except for acenaphtylene and indeno(123,cd)pyrene, which were detected by ultraviolet irradiation.

The limits of quantification (LOQ) were 0.005 mg·kg$^{-1}$ for fluorene and anthracene; 0.01 mg.kg$^{-1}$ for acenaphthene,
phenanthrene, fluoranthene, pyrene, benzo(a)anthracene, benzo(b)fluoranthene, benzo(a)pyrene, and
indeno(123,cd)pyrene; 0.02 mg·kg$^{-1}$ for naphthalene and dibenzo(ah)anthracene; 0.03 mg·kg$^{-1}$ for acenaphtylene;
and 0.05 mg·kg$^{-1}$ for chrysene, benzo(k)fluoranthene, and benzo(ghi)perylene. The complete method for PAH
analysis is referenced in the European analytical standard (NF EN 16181; AFNOR, June 2018).

Finally, the expanded uncertainty was estimated for every PAH through repeatability and reproducibility tests
using a coverage factor of two, which gives a confidence level of 95%. The results of the uncertainty calculations
reported in the supplementary material (Table S1) highlighted the fact that the acenaphtylene uncertainty was
higher than its associated concentrations; therefore, acenaphtylene was not considered in this study. In addition,
the loss of the most volatile compound Naph could have occurred due to the soil sampling technique using

composite samples and due to the soil analysis occurring years after the original sample collection. In addition, to
avoid overestimation of the total PAH content, values under the LOQ were considered to have no PAH content
(i.e. 0 µg·kg$^{-1}$).

### 2.3 Geostatistical modelling

A geostatistical modelling approach was used to map the Σ15PAHs and their probability of exceeding a selected

threshold (Webster and Oliver, 2007). The observations include many null values (30%) that correspond to values
under the LOQ. They also demonstrate extreme values due to locally distinct processes such as point pollution
from industrial sites. For zero-inflated and heavy-tailed distributions, standard geostatistical methods are not the
most appropriate because they often lead to poor estimates. In addition, these methods do not properly handle the
mapping of risks. Non-linear methods such as indicator kriging, disjunctive kriging, or conditional expectation

have been developed for such cases and have found wide acceptance in soil science (Chilès and Delfiner, 2012;
Emery, 2006; Rivoirard, 1994). In this study, the total PAHs were modelled as a transform of a normal Gaussian
field, $Z = \phi(Y)$. The spatial structure of the Gaussian field is first determined, and then maps of either the total
PAHs or the risk of exceeding a threshold are computed by conditional expectation.

In the case of continuous distribution of the original data, the anamorphosis function strictly increases and can be

inverted to assign to each observed value its Gaussian counterpart. Then, a common method for variogram
estimation can be used (e.g. to calculate the empirical/experimental variogram by the method of moments and then
to fit a model to the empirical variogram via non-linear least squares). Here, the proportion of null values is $p_0 =$
30; hence, the transformation function $\phi$ is identically equal to 0 for all Gaussian values lower than the Gaussian





quantile $y_0 = G^{-1}(p_0)$. It follows that the actual value of $Y$ is unknown where $Z = 0$ but is only constrained by

the inequality $Y < y_0$.

In such a case, two preliminary steps must be performed: determining the covariance of $Y$ and then assigning a Gaussian value to each data point at which $Z$ is null. For the first step, the zero-inflated distribution of the original data is transformed into an inflated Gaussian distribution with an inflated value equal to the quantile of the proportion of observed null values, $y_0$ (a normal score is applied, and then all values below $y_0$ are set to this value).

Then, to estimate the covariance function of the Gaussian field, the empirical variogram of the inflated Gaussian variable is computed, and the model of the Gaussian field (without inflated values) is fitted by non-linear least squares, taking into account the relationship between the covariances between the Gaussian variables and the covariances between the transformed variables. Once the structure of $Y$ is determined, the second step consists of simulating the values of the Gaussian field where the Σ15PAH value is null, honouring the spatial structure and

the inequality constraint. This is classically performed using a Gibbs sampler (Petitgas et al., 2017).

The validity of the best-fitted geostatistical model was then assessed in terms of the standardized squared prediction error (SSPE) using the results of a k-fold cross-validation. If the fitted model was a valid representation of the spatial variation of the Gaussian variable, then the sum of the squares of the errors would have an approximate $\chi^2$ distribution with a mean of 1 and a median of 0.455 (Lark, 2002).

As both the variogram model and the value of the Gaussian field at all data points are defined, the conditional expectation, also called the multi-Gaussian kriging, can be calculated to map the total PAHs. The simple kriging of the Gaussian variable is computed first, and then the conditional expectation of any transform of the Gaussian field is derived using the following Eq. (1):

$$\{\psi(Y_s)\}^{CE} = \int_{-\infty}^{+\infty} \psi\left(y_s^{SK} + \sigma_{SK}u\right)g(u)du, \tag{1}$$

where $y_s^{SK}$ and $\sigma_{SK}$ are respectively the kriging value at location $s$ (the nodes of the grid) and the standard deviation of the kriging error. The transform function $\psi$ can be either the initial anamorphosis to map the total PAHs or the indicator $y \mapsto 1_{y \geq \phi^{-1}(z_c)}$ to map the risk that the variable exceeds the threshold $z_c$. However, these estimates rely on the Gaussian values being simulated when the original values are null, and these simulated values are used for the kriging computation. Thus, the final estimate is established by the total expectation formula, where the

expectation over the Gaussian values constrained by the inequalities is computed by a Monte Carlo integration, i.e. averaging different outcomes produced by the Gibbs sampler.

The geostatistical analysis and the linear and non-linear predictors were performed using the R software 'RGeostats' package (MINES ParisTech/ARMINES 2020).

### 2.4 Health risk assessment

From a toxicological standpoint, PAHs exhibit threshold and non-threshold effects. The risks for residential populations were estimated based on lifetime exposure; for such a duration of exposure, the risks are mainly induced by non-threshold effects. For the ranges of the concentrations measured in soils, it has been verified that the threshold effects are never exceeded, even for shorter exposures, including for children. Thus, only the non-threshold effects of PAHs were considered in the context of this study. The estimation of cancer risk associated

with the exposure of individuals to potentially carcinogenic pollutants in French soils was calculated based on the equations developed by the US EPA (USEPA, 1991) and used in several studies (Cachada et al., 2016; Wang et al., 2017a; Yang et al., 2014). The incremental lifetime cancer risks (ILCRs) were calculated for each of the three



pathways: soil ingestion, dermal absorption, and soil particle inhalation. To perform the risk assessment, we converted the concentrations of PAHs measured in every soil sample into a BaP equivalent concentration (i.e.
BaP$_{eq,soil}$) using Eq. (2):

$$BaP_{eq,soil} = \sum_{i=1}^{n} PAH_i \times TEF_i, \tag{2}$$

with BaP$_{eq,soil}$ being the BaP equivalent concentration in soil in mg·kg⁻¹; PAH$_i$ is the concentration of the $i^{th}$ PAH molecule in the soil (mg·kg⁻¹); TEF$_i$ is the toxic equivalent factor for the $i^{th}$ PAH molecule (see Table S2 in the supplementary material).

We estimated the lifetime cancer risk for the residential population (children + adults) living in the area of the sampling site considering only their direct exposure to contaminated soil (without the ingestion of vegetables collected from the garden). We assumed the local homogeneity of PAHs in soils, considering the PAH concentrations in French soils measured in the current study as the 'background' exposure, and therefore focused on the long-term impacts on the population (i.e. non-threshold effects).

Table 1 presents the equations used to calculate the ILCRs for the residential population, and Table 2 details the values of parameters used in the calculations based on the risk assessment guidance of the US EPA (1991, 1996, 2020).

The total lifetime cancer risk (TLCR) was calculated by summing the three ILCRs for the residential population and outdoor workers.

**Table 1: Equations used for the health risk assessment of the residential population.**

| Pathway | Residential population |
|---|---|
| **Ingestion** | $ILCR_{ing,res} = \dfrac{C_{soil} \times IF_{soil/adj}}{AT \times 10^6} \times CSF_0$ <br> where <br> $IF_{soil/adj} = \left( \dfrac{EF_{c,res} \times ED_{c,res} \times IRS_{c,res}}{BW_c} + \dfrac{EF_{a,res} \times (ED_{a,res} - ED_{c,res}) \times IRS_{a,res}}{BW_a} \right)$ |
| **Dermal** | $ILCR_{der,res} = \dfrac{C_{soil} \times DFS_{soil/adj} \times ABS_d}{AT \times 10^6} \times \dfrac{CSF_0}{GIABS}$ <br> where <br> $DFS_{soil/adj} = \left( \dfrac{EF_{c,res} \times ED_{c,res} \times SA_{c,res} \times AF_{c,res}}{BW_c} + \dfrac{EF_{a,res} \times (ED_{a,res} - ED_{c,res}) \times SA_{a,res} \times AF_{a,res}}{BW_a} \right)$ |
| **Inhalation** | $ILCR_{inh,res} = \dfrac{C_{soil} \times EF_{a,res} \times ED_{a,res} \times ET_{a,res} \times (^1/_{PEF})}{AT} \times IUR$ |
| **Total** | $TLCR_{res} = ILCR_{ing,res} + ILCR_{der,res} + ILCR_{inh,res}$ |

**Table 2: Detailed parameters used in the incremental lifetime cancer risk assessment based on the US EPA (2020).**

| Parameter | | Units | Value |
|---|---|---|---|
| **AT** | Averaging time | d | 25550 (365×70) |
| **EF$_{c,res}$** | Exposure frequency (child) | d·y⁻¹ | 350 |
| **ED$_{c,res}$** | Exposure duration (child) | y | 6 |
| **IRS$_{c,res}$** | Soil ingestion rate (child) | mg·d⁻¹ | 200 |
| **BW$_c$** | Body weight (child) | kg | 15 |
| **EF$_{a,res}$** | Exposure frequency (adult) | d·y⁻¹ | 350 |
| **ED$_{a,res}$** | Exposure duration (adult) | y | 70 |



| | | | |
|---|---|---|---|
| **IRS$_{a,res}$** | Soil ingestion rate (adult) | mg·d$^{-1}$ | 100 |
| **BW$_a$** | Body weight (adult) | kg | 60 |
| **SA$_{c,res}$** | Surface area (child) | cm$^{-2}$ | 2373 |
| **AF$_{c,res}$** | Adherence factor (child) | mg·cm$^{-2}$ | 0.2 |
| **SA$_{a,res}$** | Surface area (adult) | cm$^{-2}$ | 6032 |
| **AF$_{a,res}$** | Adherence factor (adult) | mg·cm$^{-2}$ | 0.07 |
| **ABS$_d$** | Dermal absorption factor | - | 0.13 |
| **GIABS** | Gastrointestinal absorption factor | - | 1 |
| **PEF** | Particle emission factor | m$^3$·kg$^{-1}$ | 1.32×10$^9$ |
| **CSF$_0$** | Cancer oral slope factor | (mg·kg$^{-1}$ d$^{-1}$)$^{-1}$ | 1 |
| **IUR** | Inhalation unit risk | (mg·m$^{-3}$)$^{-1}$ | 0.6 |



## 3    Results and discussion

### 3.1 PAH contents and the influence of soil characteristics

#### 3.1.1    PAH concentrations and composition

Descriptive statistics of the dataset are presented in Table 3, showing summary statistics for each PAH molecule and for the sum of the 15 PAHs. Over the 2154 soils sampled, PAHs (at least one molecule) could be quantified in 1512 soils corresponding to 70% of the samples. Among the 1512 soils with measurable PAH concentrations,

732 contained more than 6 PAH molecules above the LOQ.

Fluoranthene and phenanthrene were quantified in 56% and 55% of the soil samples (Table 4), respectively, followed by benzo(b)fluoranthene (48%) and pyrene (46%). Conversely, acenaphthene was measured in less than 2% of the total soil samples.

**Table 3: Summary of descriptive statistics in µg·kg$^{-1}$.**

| PAH molecule | N > LOQ | N > LOQ (%) | SD/mean | Min | Q25 | Q50 | Q75 | Max | Mean |
|---|---|---|---|---|---|---|---|---|---|
| Naphthalene | 127 | 6 | 9.5 | < LOQ | < LOQ | < LOQ | < LOQ | 1030 | 2.7 |
| Acenaphthene | 38 | 2 | 13.1 | < LOQ | < LOQ | < LOQ | < LOQ | 160 | 0.3 |
| Fluorene | 231 | 11 | 6.1 | < LOQ | < LOQ | < LOQ | < LOQ | 250 | 1.4 |
| Phenanthrene | 1176 | 55 | 4.7 | < LOQ | < LOQ | 11.2 | 19.275 | 3470 | 19.0 |
| Anthracene | 174 | 8 | 10.4 | < LOQ | < LOQ | < LOQ | < LOQ | 555 | 1.9 |
| Fluoranthene | 1201 | 56 | 5.1 | < LOQ | < LOQ | 12.05 | 26 | 6080 | 30.3 |
| Pyrene | 996 | 46 | 5.1 | < LOQ | < LOQ | < LOQ | 19.575 | 4370 | 22.4 |
| Benzo(a)anthracene | 559 | 26 | 5.5 | < LOQ | < LOQ | < LOQ | 10.2 | 2180 | 11.3 |
| Chrysene | 180 | 8 | 9.0 | < LOQ | < LOQ | < LOQ | < LOQ | 4140 | 11.3 |
| Benzo(b)fluoranthene | 1039 | 48 | 3.6 | < LOQ | < LOQ | < LOQ | 18.5 | 2220 | 18.0 |
| Benzo(k)fluoranthene | 825 | 38 | < 4.9 | < LOQ | < LOQ | < LOQ | 8.0675 | 1460 | 8.2 |
| Benzo(a)pyrene | 658 | 31 | 4.7 | < LOQ | < LOQ | < LOQ | 12.3 | 1730 | 12.8 |
| Dibenzo(ah)anthracene | 148 | 7 | 8.6 | < LOQ | < LOQ | < LOQ | < LOQ | 1130 | 3.5 |
| Indeno(123cd)pyrene | 655 | 30 | 4.7 | < LOQ | < LOQ | < LOQ | 12.1 | 1830 | 10.8 |
| Benzo(ghi)perylene | 135 | 6 | 6.8 | < LOQ | < LOQ | < LOQ | < LOQ | 1530 | 7.2 |
| Σ15PAHs | 1512 | 70 | 5.1 | < LOQ[a] | < LOQ[a] | 32.6 | 121.8 | 31193 | 161.0 |
| BaPeq | 1512 | 70 | 5.2 | < LOQ[a] | < LOQ[a] | 1.13 | 17.5 | 3705 | 21.4 |
| 2–3-ring PAHs (%) | | | | NA | NA | 10.3 | 21.3 | 100 | 18.5 |
| 4-ring PAHs (%) | | | | NA | NA | 36.3 | 49.2 | 100 | 28.9 |
| 5–6-ring PAHs (%) | | | | NA | NA | 22.8 | 39.8 | 100 | 22.7 |

[a]: for the sum of the 15 PAHs, < LOQ means that no PAH molecules were quantified

The results highlight the large differences between the levels of soil contamination, with PAH concentrations varying from 5.1 to 31200 µg·kg$^{-1}$ in the Σ15PAHs. However, extreme values were not frequent, as 90% of the PAH concentrations measured were under 500 µg·kg$^{-1}$ in the Σ15PAHs, and only 4% of the concentrations were

above 1000 µg·kg$^{-1}$ in the Σ15PAHs.

The concentrations of the Σ15PAHs measured in the current study were in the same range as the PAH contents measured in rural soils from various locations in Europe and China (Table 4), which also demonstrated large variations in the Σ15PAH concentrations. The highest concentrations reported in the literature were mostly attributed to local sources, such as urban and industrial areas. Results similar to those in this study were found in

Poland with respect to the PAH distribution, with 75% of the soil samples being characterized by PAH content ≤ 694 µg·kg$^{-1}$ (Maliszewska-Kordybach et al., 2008).



**Table 4: PAH concentrations in soils reported in the literature in µg·kg⁻¹.**

| | Location | Land-use type | Concentrations in µg·kg⁻¹ | Number of PAHs considered | |
|---|---|---|---|---|---|
| | This study | | 5.1–31193 | 15 | |
| **France** | Orgeval | Rural (n = 25) and urban (n = 8) | 60–5305 | 13 | (Gateuille et al., 2014b) |
| | Rouen | Rural (forest) | 450–5650 | 14 | (Motelay-Massei et al., 2004) |
| | Paris conurbation | Rural (n = 12) and urban (n = 20) | 150–55000 | 15 | (Gaspéri et al., 2018a) |
| **Norway** | Transect to Oslo | Forest soils | 10–2600 | 16 | (Jensen et al., 2007) |
| **Poland** | Entire country | Rural (n = 217) | 80–7264 | 16 | (Maliszewska-Kordybach et al., 2008) |
| **Switzerland** | Entire country | Urban (n = 2), semi-rural (n = 13), rural (n = 7), remote (n = 1) | 19–6870 | 16 | (Gubler et al., 2015) |
| **Germany** | Entire country | Forest soils (n = 447) | 105–14889 | 16 | (Aichner et al., 2013) |
| **UK** | England, Scotland, Wales, and Northern Ireland | Rural (n = 122) | 24–128000 (mean: 608) | 16 | (Bull and Collins, 2013) |
| **China** | Entire country | Rural (n = 120) and urban (n = 28) | 9.9–5910 (mean: 377) | 16 | (Ma et al., 2015) |
| | Dongjiang River Basin | Rural (n = 30) | 23.5–231 (mean: 116) | 16 | (Zheng et al., 2014) |

### 3.1.2 Influence of soil physicochemical parameters

The PAHs measured in the samples were mostly 4- and 5–6-ring PAHs respectively representing median values of 36% and 23% of the total PAHs, with the 2–3-ring PAH contribution being around 10% (Table 4). These observations are in accordance with literature values showing a lower content of light PAHs in soils over time and a stable content of heavy PAHs (Cui et al., 2020; Gubler et al., 2015). The sorption of PAHs on organic matter due to their lipophilic properties is known to be a key factor in PAH evolution in soils (Maliszewska-Kordybach

et al., 2008; Yu et al., 2018). As the soil organic matter structure is strongly influenced by pH, the fate of PAHs could also be influenced by the soil pH, especially in forest soils (Aichner et al., 2013; Wenzel et al., 2002). Nevertheless, in the current study, no clear influence of organic carbon, organic matter (when available), or pH on the PAH content was evident, except for in urban parks (see Figures S2, S3, and S4 in the supplementary material). The low or non-existent impact of soil characteristics on the PAH concentrations found in multiple studies

(Aichner et al., 2013; Maliszewska-Kordybach et al., 2008) could be attributed to the stronger influence of other factors such as source proximity and meteorological parameters (Heywood et al., 2006; Lohmann et al., 2000). In soils receiving identical amounts of PAHs emitted by anthropogenic activities, local variations in the pH and total organic carbon might exert an influence on the fate of those contaminants in soils (Wenzel et al., 2002). This is supported by the correlation between the PAH content and soil characteristics for soils in urban parks, as urban

areas have highly concentrated PAH emissions from sources such as road traffic, industries, and residential emissions (Cachada et al., 2012b; Keyte et al., 2016). However, considering that five soils in this study were collected from urban parks, a similar conclusion cannot be made based on our dataset. At the scale of the entire





country, the influence of soil characteristics on PAH content would be minimized by the proximity of major emission sources (Aichner et al., 2013).

In addition to soil physicochemical characteristics, relationships between the PAH content in soils and land-use type have been noted in some studies (Maliszewska-Kordybach 1999); however, no differences in PAH content were observed among forests, cultivated lands, and permanent pastures in this study (see Figure S4 in the supplementary material). Only soil located in natural areas seems to present a lower content of PAHs (Kruskal-Wallis test, *p-value* < 0.05), but these results should be considered with caution since the number of natural areas

accounted for less than 5% of the 2154 sites (Section 2.1). Therefore, agricultural practices appear to have a minor influence on the PAH concentrations measured in French soils.

### 3.2 Spatial distribution of PAHs

#### 3.2.1 PAH-level distribution over France

In this study, we provide the first national map of the Σ15PAHs in Figure 2A using multi-Gaussian kriging, which

is a standard and versatile method. Once the kriging of the Gaussian variable is computed, maps for various transform functions can be easily derived. Here, it is particularly difficult to properly handle the LOQ, so a Monte Carlo integration was added for this purpose.

No evident anisotropy can be seen on the sample variogram; the omnidirectional estimated variogram and fitted model are shown in Figure 2C. The variogram displays a very large nugget effect, showing the substantial short-

range variability for the Σ15PAHs induced by the contamination process. A nested model was used to combine a nugget and a spherical model (range = 488 km). The conditional expectation approach assumes a stationary Gaussian field model. To meet this assumption, we forced the sill of the variogram model to 1, which corresponds to the total variance of the normally scored transformed variable. Notably, the estimated variogram continuously increases after 400 km, which corresponds to the north-to-south gradient observed in the data; however, this issue

could be easily solved via point-kriging with a moving 250 km radius neighbourhood and a maximum of 150 data points (Rivoirard and Romary, 2011) so that only the well-fitted part of the variogram intervenes. The results of the 10-fold cross-validation gave a mean SSPE value of 0.92 and a median SSPE value of 0.447, with both values falling within the confidence interval of the theoretical values. Finally, the R² in the transformed space equalled to 0.33.





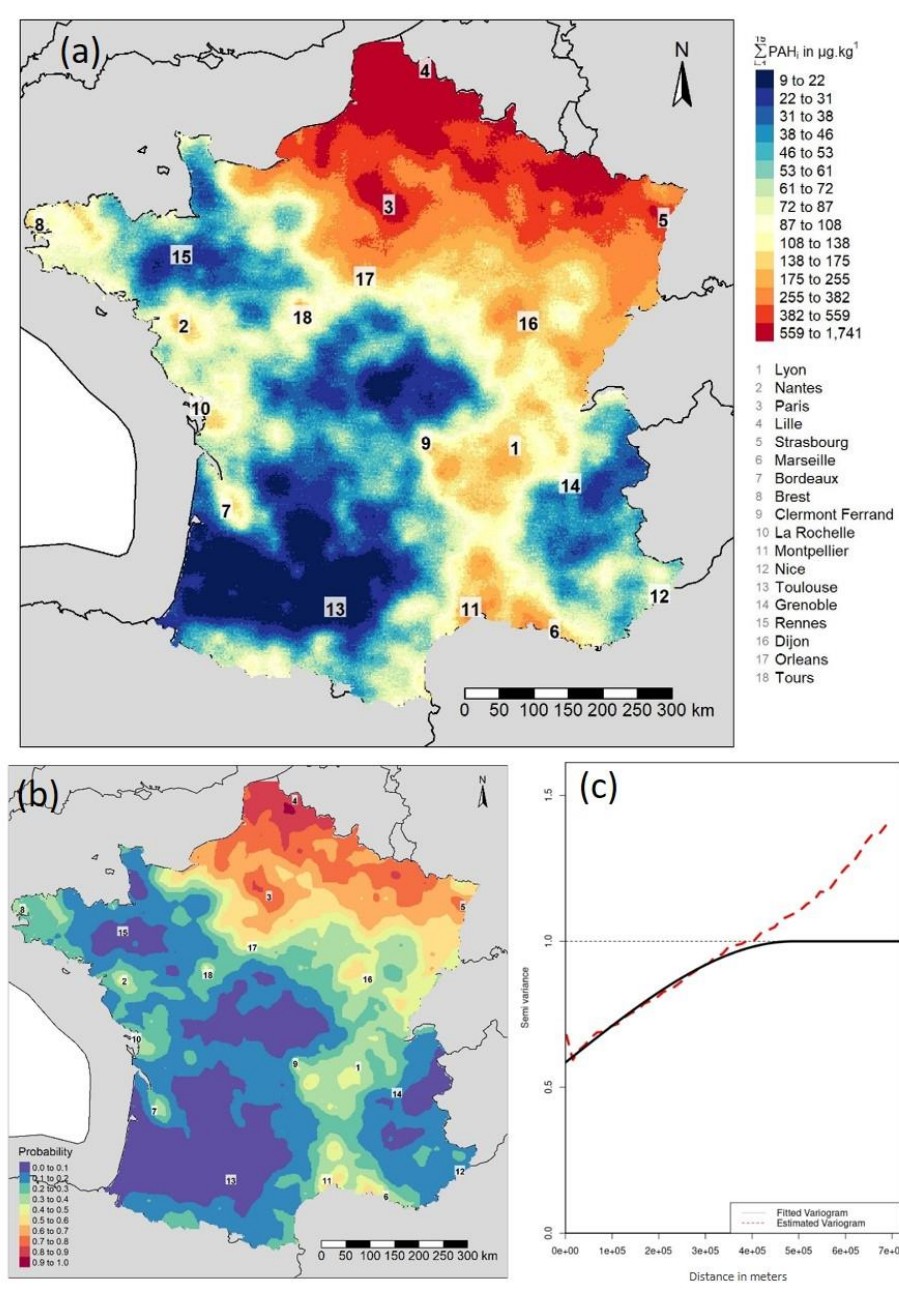

**Figure 1: (a) The estimated spatial distribution of the total PAH (Σ15PAHs) concentrations in µg·kg$^{-1}$ by conditional expectation along with major cities in France; (b) map of the estimated probability of exceeding 100 µg·kg$^{-1}$ in the Σ15PAHs; (c) estimated (in red) and fitted (in black) variograms of the normally scored transformed Σ15PAHs.**

The spatial distribution of the PAH concentrations in French soils highlighted wide variations in the contamination throughout the French territory. The north-eastern part of France, known to be an industrialised region, showed the highest levels of PAHs in the soils, as noted by Villanneau et al. (2013) for benzo(b)fluoranthene, fluoranthene,



pyrene, and phenanthrene. In addition, higher PAH concentrations could be observed in the region of the Rhône Valley, which is also known to be industrialised, and around major harbour cities along the west coast (Figure 1B). Finally, hot spots of PAHs were found near major cities such as Paris, Marseille, Lyon, Nantes or Bordeaux (Figure

1A). Those late observations contradicts the results of Villanneau et al. (2013), showing mostly the North-East – South West gradient without any evidence of cities impacts on soil contamination by PAHs.

To evaluate the state of contamination of the French soils, PAH concentrations should be compared with international historical background values, as PAHs could be emitted by natural sources such as wild fires during the Jurassic period (Killops and Massoud, 1992). Some studies have suggested a lower limit of 100 µg·kg⁻¹ in the

total PAHs, as in Canada (Canadian Council of Ministers of the Environment, 2010) or in the Seine River basin (Gaspéri et al., 2018b; Gateuille et al., 2014c). Maliszewska-Kordybach (1996) suggested a classification of agricultural soils in Poland based on the Σ15PAHs concentrations, as presented in Table 5.

**Table 5: The soil contamination classification suggested by Maliszewska-Kordybach (1996) and a design-based estimation of the proportion of associated French soils.**

| Soil classification | PAH ranges (µg·kg⁻¹) | Proportion of French soils (in %) |
|---|---|---|
| Non-contaminated | < 200 | 83 |
| Weakly contaminated | 200–600 | 12 |
| Contaminated | 600–1000 | 2.1 |
| Heavily contaminated | > 1000 | 2.9 |


The values outlined in Table 5 are supported by studies of pre-industrial periods (i.e. before 1830) such as Fernández et al. (2000), which showed PAH concentrations of 20–100 µg·kg⁻¹ in 1830 in lake sediment cores across Europe. Other studies reported PAH concentrations under 200 µg·kg⁻¹ in sediments from 1400 to 1850 (Elmquist et al., 2007; Leorri et al., 2014).

Based on the classification of Maliszewska-Kordybach (1996), the proportions of French soils attributed to each soil class were calculated (Table 5), showing that 17% could be considered contaminated when using a threshold value of 200 µg·kg⁻¹. When applying the Canadian background value of 100 µg·kg⁻¹ (Table 5), 30% of the French soils could be considered contaminated (i.e. above the threshold value). When considering the outputs of the geostatistical model (Figure 1A), 30% of the surface of France has PAH concentrations in soils higher than the

background value of 100 µg·kg⁻¹. To better assess the risk of exceeding this background value, we used a geostatistical model to map the probability that the Σ15PAHs concentration in soils exceeds 100 µg·kg⁻¹ (Figure 1B). Industrialised regions in northern France exhibit probabilities of exceeding 100 µg·kg⁻¹ above 0.8. High probabilities of soils exceeding the threshold value were also found in the vicinity of the biggest cities in France, such as Paris, Marseille, Lyon, Nantes, La Rochelle, and Bordeaux (Figure 1B).

**3.2.2    Enhancing spatial predictions**

To take into account the uncertainty of the local mean, the ordinary multi-Gaussian kriging (Emery, 2006) was tested, but there is no significant difference from the standard conditional expectation presented here, as the ordinary kriging is very close to the simple kriging due to regular sampling. The variogram model of the latent Gaussian variable presents a significant nugget effect, which may result from local contamination or measurement





errors, as previously mentioned. This introduces the issue of the support considered for the measurement (20 m ×
20 m) and for the interpolated grid (with a mesh of 2000 m × 2000 m). Taking into account that the support and
measurement errors in the Gaussian model may improve the spatial definition (see Chilès and Delfiner 2012 for a
review of models for support changes and example applications).

The standard empirical variogram averages the squared differences from different zones: the variogram model is
isotropic and stationary, but final maps suggest a north/south (or north-west/south-east) contrast and local
anisotropies, which may or may not be partially explained by external co-variates (e.g. topography, distance to the
sea, and dominant wind direction). Hence, modelling local anisotropies (Fouedjio et al., 2016) or using a non-
stationary SPDE model (Pereira, 2019) could improve the structured part of the variogram model and, hence, the
maps.

**3.3  Major origin of PAHs using molecular diagnostic ratios**

To identify the origin of PAHs in French soils, the four widely used molecular diagnostic ratios—Flh/(Flh + Py),
Ant/(Ant + Phe), BaA/(BaA + Chry), and IdP/(IdP + BghiP)—were investigated to distinguish among petrogenic
(i.e. crude oil), pyrogenic (coal, wood, and biomass combustion), and fossil fuel combustion sources (Table 6).
We computed these ratios and mapped the results using a simple point map (Figure 2) without any interpolation,
and missing values corresponded to sites at which none of the PAHs used in the ratios were quantified.

**Table 6: Most commonly used molecular diagnostic ratios associated with PAH origins.**

| Ratio | Petrogenic | Fossil fuel combustion | Pyrogenic | References |
|---|---|---|---|---|
| **Flh/(Flh + Py)** | 0–0.4 | 0.4–0.5 | 0.5–1 | |
| **Ant/(Ant + Phe)** | 0–0.1 | | 0.1–1 | (Biache et al., 2014; Ravindra et al., 2008; |
| **BaA/(BaA + Chry)** | 0–0.2 | 0.2–0.35 | 0.35–1 | Tobiszewski and Namieśnik, 2012; Yunker et al., 2002; Zhang et al., 2005) |
| **IdP(IdP + BghiP)** | 0–0.2 | 0.2–0.5 | 0.5–1 | |



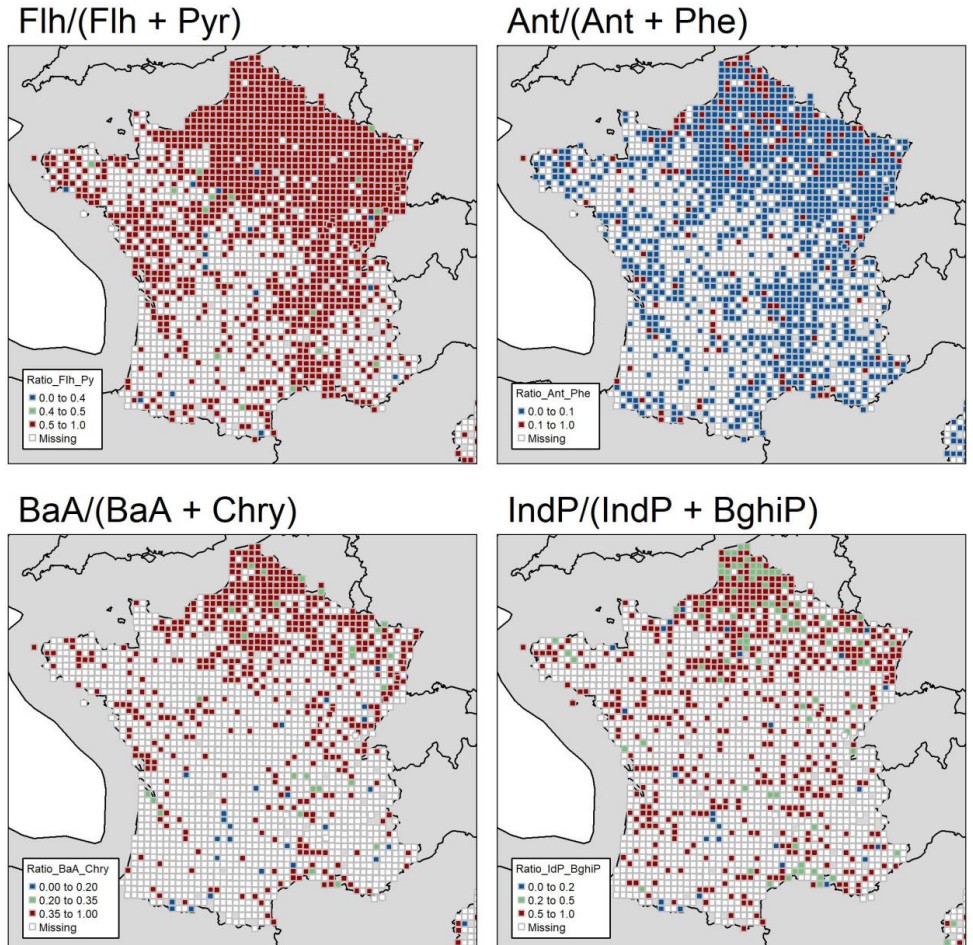

**Figure 2: Spatialized molecular diagnostic ratios in French soils. Missing values correspond to sites at which none of the PAHs used in the ratios were quantified.**


The spatial distribution of the four diagnostic ratios appears to be similar to the majority of calculated ratios in the northern part of France (Figure 2) but with different indicators with respect to the PAH origins. The values of the Flh/(Flh + Py), IdP/(IdP + BghiP), and BaA/(BaA + Chry) ratios are mainly associated with pyrogenic signatures attributed to coal, wood, and biomass burning (Figure 2A, C, D), whereas the Ant/(Ant + Phe) values indicate

petrogenic signatures attributed to crude oil (Figure 2B). This difference in the PAH origins based on the Ant/(Ant + Phe) ratio could be explained by the sensitivity of this ratio to environmental processes such as oxidation or biodegradation (Katsoyiannis and Breivik, 2014; Tobiszewski and Namieśnik, 2012). Phenanthrene and anthracene are lighter PAHs susceptible to multiple processes during their transport, shifting the source signature of Ant/(Ant + Phe) (Biache et al., 2014; Zhang et al., 2005). The lower detection of anthracene (8%) compared to

phenanthrene (55%), resulting in many instances of Ant/(Ant + Phe) being close to 0, probably reflects anthracene



depletion rather than PAH origins. Consequently, Ant/(Ant + Phe) should be used with caution when trying to identify the origin of PAHs and was thus considered an unreliable source signature in this study.

As shown by the Flh/(Flh + Py), IdP/(IdP + BghiP), and BaA/(BaA + Chry) diagnostic ratios (Figure 2A, C, and D), pyrogenic combustion such as coal and wood burning might be the main origin of PAHs measured in the soil

of the north-eastern part of France and the Rhône Valley in the south-east. In the literature, Flh/(Flh + Py) and IdP/(IdP + BghiP) have been identified as the most conservative ratios due to the lower sensitivity of Flh, Py, IdP, and BghiP to degradation (Brändli et al., 2008; Tobiszewski and Namieśnik, 2012). In the case of BaA/(BaA + Chry), shifts in the signature have been observed during atmospheric transport due to photo-degradation (Katsoyiannis and Breivik, 2014); however, the concordance of the latter ratio with the PAH origins identified

using the Flh/(Flh + Py) and IdP/(IdP + BghiP) diagnostic ratios suggests that it is a reliable molecular diagnostic ratio in this study.

In addition to the indication of pyrogenic origins of the PAHs by molecular diagnostic ratios, the homogeneous spatial pattern of both PAH concentrations and the ratio in north-eastern France (Figure 1 and 2) suggest contamination through long-range atmospheric deposition, the latter being the major source of soil PAHs

(Tobiszewski and Namieśnik, 2012). Multiple studies reported high PAH deposition rates in Western Europe from 1900 to 1970 due to industrial development, especially from heavy industries (e.g. coal and steel), with a massive peak in 1960 before a decrease due to pollution control (Fernández et al., 2000; Gabrieli et al., 2010; Leorri et al., 2014). An estimation of the BaP emissions in Europe by Pacyna et al. (2003) identified Germany, France, and the UK as the main contributors, emitting 22%, 8%, and 11% of the 1300 tons of BaP released in 1970, respectively.

In addition, the Gabrieli et al. (2010) study of atmospheric PAH deposition beginning in 1700 showed that, during 1900–1960, the Flh/(Flh + Py) ratio decreased from 0.65 to 0.55. This decreasing trend was explained by the increasing contribution of fossil fuel combustion (gasoline, diesel) to local atmospheric PAH deposition (Gabrieli et al., 2010). This value of 0.6 in Flh/(Flh + Py) appears to be the main value in the north-eastern part of France (Figure 2A), therefore supporting historical industrial PAH emissions as a major source of PAHs in the soils of

this region. This hypothesis of the coal industry's contribution to PAH loads in Northern France during the mid-20[th] century was also suggested by Lorgeoux et al. (2016) using organic pollutant records from a sediment core of the Seine estuary.

The persistence of PAHs over more than a century and the absence of changes in the signature could be explained by multiple factors. First, the amount of PAHs emitted at the beginning of the 20[th] century was 10 to 100 times

higher than the current PAH depositions (Fernández et al., 2000; Gabrieli et al., 2010). The high rates of PAH deposition during those 30 to 60 years could have prevented any changes in the PAH signatures in soils associated with source modifications after 1960 due to the stock of PAHs already polluting soils (Froger et al., 2019; Gateuille et al., 2014b). Second, the process of PAH sequestration in the soil could explain their long-term persistence in French soils (Biache et al., 2011). This sequestration is caused by the sorption of PAHs on organic matter and their

diffusion through micropores in soils, reducing their bioavailability for microorganisms (Bogan and Sullivan, 2003; Semple et al., 2003). Overall, the aging of soils could increase PAH interactions in soils, enhancing the slow but irreversible sequestration of PAHs and reducing their potential degradation in soils (Anyanwu and Semple, 2015; Duan et al., 2015; Ma et al., 2012; Ouvrard et al., 2013).

Additionally, the hotspots of PAH concentrations around major cities (Figure 1) suggests local sources of

contaminants, such as in urban or suburban environments, and multiple sources of PAHs could contribute to PAH





soil contamination. Current sources of PAHs, such as road traffic, local industries, and residential emissions (household heating) (Cachada et al., 2012a; Gateuille et al., 2014a; Lohmann et al., 2000) as well as legacy contamination due to former industries (Bertrand et al., 2015; Pies et al., 2007) could contribute to soil PAH content. However, precise source identification and apportionment cannot be done based only on PAH molecular

diagnostic ratios, which only provide information about pyrogenic, petrogenic, or mixed PAHs in soils (Brändli et al., 2008; Tobiszewski and Namieśnik, 2012; Yunker et al., 2014). Multivariate analysis associated with linear mixing models and positive matrix factorization could be performed to reveal PAH sources and their respective contributions to PAH contamination in soils (Harrison et al., 1996; Larsen and Baker, 2003; Wang et al., 2015). Additionally, each PAH molecule may present different spatial structures depending on its source. Minimum and

maximum auto-correlation factors (Petitgas et al., 2018; Switzer and Green, 1984) deduced from different indicators (for example, the indicators of the $1_{PAHi} \geq LOC_{LOQ}$) could be used. These factors are the spatial counterpart of the standard PCA, and their mapping could help regionalize the origin of PAHs and estimate the risk using alternate non-linear methods such as discrete disjunctive kriging.

### 3.4  Health risk assessment

#### 3.4.1    Total lifetime cancer risks in France

The results of the health risk assessment evaluated for the residential population are presented in Table 7. According to the US Environmental Protection Agency, the elevated risk is attributed to a TLCR above $10^{-4}$, meaning that the probability of an individual developing cancer over a lifetime is more than one in 10000. A TLCR value under $10^{-6}$ is considered to indicate virtual safety, meaning that the probability of an individual developing

cancer is less than one in 1000000. Consequently, TLCRs between $10^{-4}$ and $10^{-6}$ were associated with a 'moderate risk'. However, the limit of $10^{-6}$ is commonly used by the US EPA as the target risk value to calculate screening levels in soils. In France, the ministerial circular on December 10[th], 1999 chose a value of $10^{-5}$ as the limit defining the sites needing remediation for polluted soils.

Among the 2154 soils collected throughout the French territory, 54% had a TLCR value (i.e. 1512 sites), indicating

that those soils were contaminated by at least one of the seven carcinogenic PAHs. TLCR values for the residential population ranged from $1.89 \times 10^{-11}$ to $1.37 \times 10^{-5}$, with 20 sites presenting TLCRs above the safety target value of $10^{-6}$ (i.e. 1% of the sites). Only one site showed a TLCR value above $10^{-5}$, which was considered a polluted site according to French legislation. Ingestion of soil contributed to 68.8% of the total cancer risk compared to dermal contact (31.2%) and inhalation (0.01%), which is commonly observed in the literature (Tong et al., 2018; Wang

et al., 2017b).

Considering that soil samples were collected mainly in locations distant from anthropogenic sources (both industries and urban areas), the current study could be considered an evaluation of the background level of PAHs in French soils. Consequently, soils contaminated by PAHs showed low risks for residential populations, considering that 99% of the soils presented TLCR values under $10^{-6}$ (Table 7). In the literature, higher TLCR

values have been identified in southern Italy with mean ILCRs of $4.77 \times 10^{-6}$ for adults (Qi et al., 2020) and in China with a mean TLCR value of $1.86 \times 10^{-6}$ in Shanghai (Tong et al., 2018) and an ILCR value of $4.4 \times 10^{-5}$ in the Yangtze River Delta (Wang et al., 2017b). However, those studies were conducted mainly on industrial and highly urbanised regions and reflect the risks induced by soils under high anthropogenic pressure and not the background contamination over a large territory.





**Table 7: Summary of the total lifetime cancer risk (TLCR) attributed to French soils for the residential population.**

|  | **TLCR residential population** |
|---|---|
| **min** | $1.89 \times 10^{-11}$ |
| **max** | $1.37 \times 10^{-5}$ |
| **mean** | $1.13 \times 10^{-7}$ |
| **q25** | $3.82 \times 10^{-9}$ |
| **q50** | $1.07 \times 10^{-8}$ |
| **q75** | $9.63 \times 10^{-8}$ |
| **Number of sites < $10^{-7}$** | 1144 |
| **Number of sites between $10^{-7}$ and $10^{-6}$** | 348 |
| **Number of sites > $10^{-6}$** | 20 |

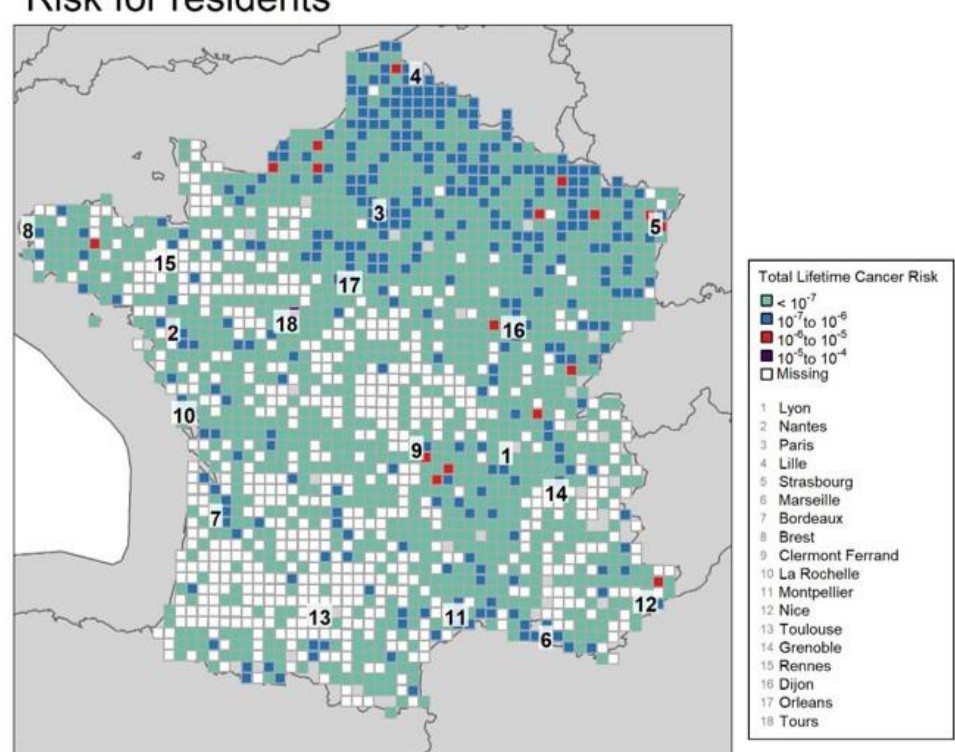

**Figure 3: Health risk assessment for the residential population showing the total lifetime cancer risk (TLCR) for French soils along with the location of major cities. Missing values correspond to sites at which no PAHs were quantified, so a TLCR below $10^{-7}$ can be assumed.**

**Erreur ! Source du renvoi introuvable.** presents the residential maps of TLCRs and indicates that the north-eastern part of France and the Rhône Valley had the most sites with estimated TLCRs reflecting the national PAH concentration distribution (Figure 1A). These observations can be explained by the positive correlation between the BaP toxic equivalent and the Σ15PAHs shown in Figure 4. These results indicate that the contamination of





soils by PAHs is driven by the most carcinogenic PAHs with the highest TEFs (i.e., BaA, Chr, BbF, BkF, BaP,
IndP, and DahA), whose concentrations are proportional to the total PAHs measured.

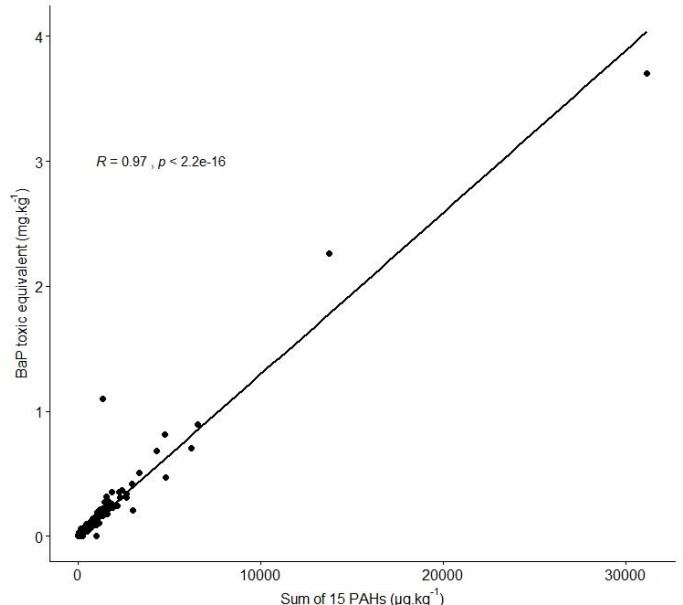

**Figure 4: Correlation between the BaPeq in mg·kg⁻¹ and the concentration of the Σ15PAHs in µg·kg⁻¹ in French soils.**

The soils with TLCRs between $10^{-7}$ and $10^{-6}$ are concentrated in the northern part of France and in the vicinity of
the main cities (**Erreur ! Source du renvoi introuvable.**). Among the 17 main cities shown in Figure 3, 11 are
located in areas in which TLCRs could be calculated, and four are situated in areas with multiple sites with TLCRs
between $10^{-7}$ and $10^{-6}$. As multiple sources of PAHs are concentrated in urban areas, soils with higher
contamination levels that are usually found in cities might pose higher risks for the population, as in Naples, where
the ILCRs were above $1 \times 10^{-5}$ (Albanese et al., 2015). In France, a database of 300 urban soils was created
(ADEME, 2018), and the soils therein had a median value of 1560 µg·kg⁻¹ in the Σ16PAHs, which is 50 times
higher than the 32 µg·kg⁻¹ in the Σ15PAHs in the current study. Therefore, the higher risks posed by PAHs in soils
would be expected in urbanised regions. In addition, the distribution of soils with higher TLCRs around cities
(especially Paris, Lille, and Marseille) highlights the influence of densely urbanised areas on the surrounding soils
over a distance of up to several dozen kilometres (Jensen et al., 2007). Finally, the 20 sites with the highest TLCRs
above $10^{-6}$ appeared to be randomly distributed throughout the territory, suggesting local sources of PAHs at those
sites, such as heavy industries.

The first results of the background national health risks linked to PAHs show the spatial heterogeneity in the risks
encountered by the population and the necessity of considering soils in risk evaluations and the national
surveillance of contaminants in the environment.

### 3.4.2    Limits and perspectives

As we used a site-specific determinist method to predict the TLCR at each site, the variations in the population
characteristics and model parameters were not integrated in the calculation. Therefore, the use of a probabilistic



model with a sensitivity analysis would result in more accurate predictions of cancer risks based on French population statistics (US EPA 2001; Bruce et al. 2007; Tong et al. 2018). However, the use of the point estimate approach is the first step recommended by the US EPA (2001) when conducting risk assessments. These results

provided an overview of the health risks associated with background PAH concentrations in French soils and the need to further evaluate these risks.

Three pathways of human exposure to PAHs in the soils were considered (i.e. ingestion, dermal contact, and inhalation). To integrate all exposure routes, PAH transference from soil to food production (crops, vegetables, and fruits) should be taken into account as well as groundwater transfer. Many studies have demonstrated that

exposure through food ingestion significantly contributes to the total risk, as in the Yangtze River Delta, where TLCRs rose from $4.4 \times 10^{-5}$ to $9.17 \times 10^{-4}$ (Wang et al., 2017b). Similarly, in southern Italy, the risk induced by food ingestion exceeded the risks of the other pathways by two orders of magnitude (Qi et al., 2020). Therefore, the potential risks posed by PAHs in French soils could be underestimated, especially since a significant part of the production of cereals (soft wheat and barley) and some vegetables (potato) are cultivated in northern France.

Consequently, a more in-depth health risk study integrating land-use types (crops, grassland, and forest), produce cultivation, and a scenario in which the residential population raises their own vegetables would result in deeper insight into the risks induced by PAHs in soils.

The risk estimations were made based on the additional toxic effects of PAHs using toxic equivalent factors (TEFs). However, little is known about the potential synergistic, antagonist, or additive effects of PAH mixtures

on the human body, which could lead to either underestimation or overestimation of ILCRs. In addition, the proportion of PAHs absorbed in the body resulting in harmful effects is considered to be 100%; however, Hack and Selenka (1996) identified the influence of food type on the mobilisation of PAHs from soil in the digestive tract. Therefore, the population's diet could also influence the potential risks associated with soil contaminants. Consequently, to better evaluate the risk induced by food ingestion, bioavailability tests should be conducted using

the standard diet of the targeted population. Furthermore, as PAHs in French soils were identified as originating from historical contamination, the aging process might reduce PAH bioavailability (Bogan and Sullivan, 2003; Semple et al., 2003) as well as their potential toxicity to organisms (Duan et al., 2015; Ma et al., 2012). Nevertheless, Reeves et al. (2001) demonstrated the low impact of soil aging on PAH availability and ingestion by organisms (i.e. rats in the study).

Finally, the sequestration of PAHs in the soil could result in the biodegradation of some molecules into metabolites such as fluoranthene-2,3-dione(F23Q), which is known to be toxic, can accumulate in the environment, and inhibits the degradation of other PAHs (Kazunga et al., 2001). This issue was also reported by Schmidt et al. (2010), who demonstrated that fungal PAH-metabolites tend to accumulate in the soil solution due to their high water solubility and appear to mineralise more slowly than the parent PAHs potentially leaching into the

groundwater. These results support the need to study the fate of contaminants in soils and their associated risks.

## 4   Conclusion

This study explored the concentrations of PAHs in 2154 soils collected throughout France based on an extended dataset collected using a systematic sampling grid covering all land-use types. Seventy percent of the soil samples presented PAH concentrations over the LOQ, and 30% of those soils were above the background value of 100 µg

kg$^{-1}$ in the Σ15PAHs, with a probability of 0.8 to exceed this threshold value in northern France. The spatial



distribution of the $\Sigma$15PAHs throughout the country showed a particular pattern, with higher PAH content in soils situated in the north-eastern part of France, in the Rhône Valley, and near the main cities. These results demonstrated large-scale contamination of French soils by PAHs, which was identified by PAH molecular ratios as legacy contamination from the emissions linked to the industrialisation of Europe that started in 1850. The

health risk assessment conducted for the residential population showed that 99% of the French soils presented a risk lower than the target value of TLCR of $10^{-6}$ for the residential population. Only one site exceeded the limit of $10^{-5}$ chosen by the French government to consider soil as contaminated in residential areas. A higher density of TLCRs between $10^{-7}$ and $10^{-6}$ was found in the north-eastern part of France, the Rhône Valley, and in soils surrounding the main cities. These results demonstrate the need to investigate the risk associated with PAHs in

soils using a probabilistic method with French population statistics that includes food ingestion as a transfer pattern. Finally, this study demonstrated the heterogeneous risks associated with PAHs in soils over France; highlighted the environmental inequalities with regional variations in population and ecosystem exposure to hazardous contaminants; and indicated the need to conduct large-scale studies to understand contaminant behaviours, their fates in the ecosystem, and their associated risks.


**Credit Author Statement**

Claire Froger: Conceptualization, Data Curation, Writing – Original Draft. N. P. A. Saby: Conceptualization, Supervision, Methodology, Data Curation, Formal Analysis, Writing – Review & Editing. C. C. Jolivet: Supervision, Resources, Writing – Review & Editing. L. Boulonne: Supervision, Resources, Writing – Review &

Editing. G. Caria: Resources, Methodology, Validation, Writing – Review & Editing. X. Freulon: Software, Formal Analysis, Validation, Writing – Review & Editing. C. de Fouquet: Software, Formal Analysis, Validation, Writing – Review & Editing. H. Roussel: Methodology, Validation, Writing – Review & Editing. F. Marot: Methodology, Validation, Writing – Review & Editing. A. Bispo: Supervision, Writing – Review & Editing.

**Competing interests**

The authors declare that they have no conflict of interest.

**Acknowledgements**

The soil sampling and the analyses of physicochemical properties of soils were supported by the French Scientific

Group of Interest on soils: the ''GIS Sol,'' involving the French Ministry in charge of the Ecological Transition (MTE), the French Ministry in charge of the Agriculture and Food (MAA), the French Agency for Environment and Energy Management (ADEME), and the National Institute for Agronomic and Environmental Research (INRAE). We thank all the soil surveyors and technical assistants involved in sampling the sites. The PAH analyses were supported by a grant from the French Observatory for the Pesticides Residues (ORP).




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
