# Peer review of "Spatial variations, origins, and risk assessments of polycyclic aromatic hydrocarbons in French soils"

_SOIL, 2021_

## Author Comment (AC1)

Please consider adding the information on QA/QC, namely which reference material(s) were used for PAHs

We added the values of the reference material used to validate the results in the Supplementary Material as Table S1. A sentence about the reference material has been added lines 136 to 137.

Figure S1 and S6 – please consider adding geographical coordinates to the map

Figure 1-4: the same as above

Longitude and latitude has been added to the all the maps (Figure 1 to 4, S1 and S6).

Line 104: you mentioned 16 PAHs measured, however you discuss only 15 of them (sum of 15) in your work. Please clarify better in the text why you only used 15 PAHs for discussion of the results

Sixteen PAHs were measured in soils. However, concentrations of acenaphtylene could not be validated given the value of 0 mg.kg$^{-1}$ for this compound in the reference material. Therefore, this PAH compound was not considered in the paper resulting in the use of 15PAHs in the discussion. Nevertheless, acenaphtylene was quantified in less than 1% of the 2154 soil samples and do not significantly modify the results. This explanation was added to the manuscript lines 137 to 140.

---

## Author Comment (AC2)

**General Comments:**

In the introduction, it would be better for the authors to considered the occurrence of PAHs in other environmental media, such as atmosphere and rivers, so as to explain whether soil is the "source" or "sink" of PAHs.

In the introduction we added references pointing the role of soils in the dynamic of PAHs in the environment from line 39 to 45 with literature studies at large scale showing that soils act as both "sink" and "source" of PAHs.

In the Materials and Methods section, QA/QC procedures should be performed.

The use of a reference material has been specified lines 135 to 137 and the values of this reference material has been added in the supplementary material as Table S1.

The specific source period of PAHs in soil needs to be demonstrated scientifically, such as specific period of PAHs generation could be determined by C isotope method.

Unfortunately, analyses of C isotope was not done during soil analysis procedure. However, multiple references in France and Europe support the hypothesis of a massive soil contamination by PAHs during the 20th century linked to coal consumption and industrial development. Nevertheless, your suggestion has been added as a perspective to better identify PAHs sources lines 419 to 422.

Format of the text should be modified and all Figures should be marked with longitude and latitude.

All figures have been modified with longitude and latitude. We used the guidelines from SOIL journal for the manuscript format.

It would be better for the authors to consider and quantify alkyl-PAHs from all of their samples. May be it would be more comprehensive for this work.

The goal of the project was to analyse PAHs to be able to map the contamination of French soils at the country scale. Given the high number of samples to analyse and due to financial and logistical reasons, the choice was made to focus on the 15 PAHs. However, measuring alkyl-PAHs would have been interesting to have a supplementary tool for source identification. This was added to the perspective of the work lines 419 to 422.

**Specific comments:**

Reasons why "The results of the uncertainty calculations reported in the supplementary material (Table S1) highlighted the fact that the acenaphtylene uncertainty was higher than its associated concentrations; therefore, acenaphtylene was not considered in this study." should be explained in details.

Acenaphtylene could not be validated by the reference material, therefore it was not considered in the study, this was specified lines 137 to 140. However, this compound quantified in less than 1% of the samples would not have modified the results of the study.

In section 3.3, it was mentioned that the concentration and ratio of PAHs indicated that pollutants were caused by distant atmospheric deposition. Relevant literature cited should be given.

Line 380, "long-range emissions" was modified to "regional scale emissions". The homogeneous pattern of PAHs concentrations and ratios in French soils suggest regional emissions of PAHs more than emissions by localized industries. This is supported by many literature studies showing very high PAHs emissions in Northern Europe during the 20th century, cited from line 381 to 392.

Figure 2, the ratio points involved in the figure are mainly concentrated in the north, which contains a large number of missing values and reduces the representativeness of the sampling points.

Missing values pointed out in the Figure 2 correspond indeed to samples where both compounds used to calculate the ratios were not measured. However, these missing values are mostly distributed in the South West of the country were lower PAHs concentrations were measured and therefore where the contamination is very low. Moreover, in the northern part of France, the presence of missing values varied among the ratios considered. For Flh/(Flh + Pyr), only a few number of samples exhibited missing values, which makes this ratio still worth to study and was then the most discussed in this work. Despite missing values for the other ratios, we think that their comparison to Flh/(Flh+Pyr) was useful to validate the interpretation of PAHs source. A sentence was added in the manuscript, lines 374 to 377.

Figure 4, the "R" in the figure should be changed to "$R^2$", and "p<2.2e-16 " means " $p$<0.05"?

The Figure 4 was modified.

What means for "Erreur! Source du renvoi introuvable! "in line 440 and line449.
Citation: https://doi.org/10.5194/soil-2021-6-RC2

This error referred to the Figure 3 and was corrected.